# SARS-CoV-2 antibodies and breakthrough infections in the Virus Watch cohort

Robert W. Aldridge [1] ✉, Alexei Yavlinsky[1], Vincent Nguyen[1,2], Max T. Eyre[3,4], Madhumita Shrotri[1], Annalan M. D. Navaratnam[1,2], Sarah Beale[1,2], Isobel Braithwaite[1], Thomas Byrne[1], Jana Kovar [2], Ellen Fragaszy[1,5], Wing Lam Erica Fong[1], Cyril Geismar [1,2], Parth Patel [1], Alison Rodger[6], Anne M. Johnson[6] & Andrew Hayward[3]

A range of studies globally demonstrate that the effectiveness of SARS-CoV-2 vaccines wane over time, but the total effect of anti-S antibody levels on risk of SARS-CoV-2 infection and whether this varies by vaccine type is not well understood. Here we show that anti-S levels peak three to four weeks following the second dose of vaccine and the geometric mean of the samples is nine fold higher for BNT162b2 than ChAdOx1. Increasing anti-S levels are associated with a reduced risk of SARS-CoV-2 infection (Hazard Ratio 0.85; 95%CIs: 0.79-0.92). We do not find evidence that this antibody relationship with risk of infection varies by second dose vaccine type (BNT162b2 vs. ChAdOx1). In keeping with our anti-S antibody data, we find that people vaccinated with ChAdOx1 had 1.64 times the odds (95% confidence interval 1.45-1.85) of a breakthrough infection compared to BNT162b2. We anticipate our findings to be useful in the estimation of the protective effect of anti-S levels on risk of infection due to Delta. Our findings provide evidence about the relationship between antibody levels and protection for different vaccines and will support decisions on optimising the timing of booster vaccinations and identifying individuals who should be prioritised for booster vaccination, including those who are older, clinically extremely vulnerable, or received ChAdOx1 as their primary course. Our finding that risk of infection by anti-S level does not interact with vaccine type, but that individuals vaccinated with ChAdOx1 were at higher risk of infection, provides additional support for the use of using anti-S levels for estimating vaccine efficacy.

Vaccines based on the spike glycoprotein of SARS-CoV-2 are estimated to have prevented the deaths of hundreds of thousands of people globally[1–3]. The majority of people aged over 18 in England and Wales were vaccinated with either BNT162b2 (BioNTech/Pfizer) or ChAdOx1 nCoV-19 (AstraZeneca/Oxford)[4]. We have previously reported[5] an analysis of waning SARS-CoV-2 antibodies targeting the spike protein (anti-S) after a second dose of BNT162b2 or ChAdOx1 in 605 adults who were seronegative to SARS-CoV-2 anti-Nucleocapsid (anti-N). These data suggested higher peak levels and faster waning of anti-S levels after a second dose of BNT162b2 compared to ChAdOx1 in infection-

[1]Centre for Public Health Data Science, Institute of Health Informatics, University College London, London, UK. [2]Institute of Epidemiology and Health Care, University College London, London, UK. [3]Centre of Health Informatics, Computing and Statistics, Lancaster Medical School, Lancaster University, Lancaster, UK. [4]Liverpool School of Tropical Medicine, Liverpool, UK. [5]Department of Infectious Disease Epidemiology, London School of Hygiene and Tropical Medicine, London, UK. [6]Institute for Global Health, University College London, London, UK. ✉e-mail: r.aldridge@ucl.ac.uk

naive individuals over a 3–10-week period for samples collected on June 14–15 2021.

Analysis by the UK Health Security Agency (UKHSA)[6] estimated that 20 weeks following the second dose, vaccine effectiveness for ChAdOx1 against infection was 47.3% (95% CIs: 45.0–49.6) compared to 69.7% (95% CIs: 68.7–70.5) for BNT162b2. For hospitalisations due to COVID-19, the corresponding vaccine effectiveness was 77.0% (70.3–82.3) for ChadOx1 and 92.7% (90.3–94.6) for BNT162b2. For deaths, vaccine effectiveness at 20 weeks was 78.7 (95% CI 52.7 to 90.4) for ChadOx1 and 90.4 (95% CI 85.1 to 93.8) for BNT162b2. Overall, these data suggest faster waning of protection against infection and severe disease for ChAdOx1 compared to BNT162b2 over these longer time periods.

Given the high levels of vaccination in the UK and consequently the high level of cases and deaths occurring in double-vaccinated individuals, it is important to understand trajectories of anti-S waning in individuals post second dose of COVID-19 vaccine and its association with levels of protection for SARS-CoV-2 infection, to enable serosurveillance data to inform vaccine policies in the UK and around the globe. Previous studies have examined antibody responses and levels of protection in the general population after two doses of the ChAdOx1 or BNT162b2 vaccines using data from randomised controlled trials and observational studies[7–9].

## Table 1 | Demographic and clinical characteristics of included individuals and samples

| | Samples | Individuals |
|---|---|---|
| All | 24,997 | 9492 |
| **Age group** | | |
| 18–64 | 13,329 (53%) | 4998 (53%) |
| 65+ | 11,668 (47%) | 4494 (47%) |
| **Sex** | | |
| Male | 10,476 (42%) | 4005 (42%) |
| Female | 14,521 (58%) | 5487 (58%) |
| **Ethnicity** | | |
| White British or Irish | 23,305 (93%) | 8812 (93%) |
| White Other | 1043 (4.2%) | 415 (4.4%) |
| Mixed | 154 (0.6%) | 66 (0.7%) |
| South Asian | 210 (0.8%) | 89 (0.9%) |
| Other Asian | 142 (0.6%) | 52 (0.5%) |
| Black | 54 (0.2%) | 21 (0.2%) |
| Other/Missing | 89 (0.4%) | 37 (0.4%) |
| **Clinical vulnerability** | | |
| Clinically extremely vulnerable | 3372 (13%) | 1292 (14%) |
| Clinically vulnerable | 6729 (27%) | 2572 (27%) |
| Not clinically vulnerable | 14,896 (60%) | 5628 (59%) |
| **Vaccine type (First Dose)** | | |
| BNT162b2 | 8319 (33%) | 3363 (35%) |
| ChAdOx1 | 16,054 (64%) | 5834 (61%) |
| mRNA-1273 | 183 (0.7%) | 71 (0.7%) |
| Other/Missing | 441 (1.8%) | 224 (2.4%) |
| **Vaccine type (Second Dose)** | | |
| BNT162b2 | 8399 (34%) | 3407 (36%) |
| ChAdOx1 | 16,288 (65%) | 5960 (63%) |
| mRNA-1273 | 171 (0.7%) | 66 (0.7%) |
| Other/Missing | 139 (0.6%) | 59 (0.6%) |

See Supplementary Tables S2, S3 for further details of clinically vulnerable and clinically extremely vulnerable groups.

Our study aims to build upon this prior evidence and directly compare anti-S levels and the effectiveness of BNT162b2 and ChAdOx1 in preventing SARS-CoV-2 infection in the Virus Watch cohort which enables large scale population follow-up with repeated testing of antibodies over time in a population with detailed demographic and clinical characterisation and long term follow-up. To achieve this aim we examine three research questions. First, how does anti-S waning vary after the second dose by vaccine type, demographic and clinical characteristics. Second, what is the total effect of anti-S level on risk of SARS-CoV-2 infection in England and Wales. Third, does the second dose vaccine type affect the chances of developing SARS-CoV-2 infection after receiving two vaccine doses.

## Results

### Anti-S waning by vaccine, demographic and clinical characteristics

24997 samples from 9492 N-seronegative individuals were included in the analysis of anti-S waning over time (Table 1). 5960 people (63%; 5960/9492) who received a second dose of ChAdOx1 and 3407 who received a second dose BNT162b2 (36%; 3407/9492) were included in our analyses of the trajectories of anti-S waning after the second dose by vaccine type. 192 individuals (2%; 192/9492) had a different first and second dose vaccine type. For both vaccines, waning anti-S levels followed a mean log-linear decline from 4 weeks after the second dose of vaccination (Fig. 1). The geometric mean of the anti-S samples peaked at 3 weeks after the second dose of vaccine BNT162b2 at 10555 (95% CI: 9291–11992) U/ml and at 4 weeks for ChAdOx1 at 1069 (95% CI: 925–1236) U/ml. At 20 weeks after the second dose of vaccine, the mean anti-S levels were 1611 (95% CI: 1515–1712) U/ml for BNT162b2 and 387 (95% CI: 363–414) U/ml for ChAdOx1. There was evidence that rates of waning were higher in BNT162b2 ($-6.55e{-}03$ [ln(anti-S U/ml)/day], $t_{1/2} = 72.1$ days) than ChAdOx1 ($-9.62e{-}03$ [ln(anti-S U/ml)/day], $t_{1/2} = 105.9$ days; $p < 0.001$). Twenty weeks after the second dose of vaccine, mean BNT162b2 anti-S levels were 1875 (95% CI: 1686–2085) U/ml in 18–64 year olds and 1479 (95% CI: 1373–1593) U/ml in participants over 65 years of age (Fig. 2). Anti-S levels at 20 weeks post second dose of ChAdOx1 were 420 (95% CI: 380–464) U/ml in 18–64 year olds and 356 (95% CI: 328–390) U/ml in participants over 65 years of age. There was no evidence of a difference in the rates of waning by age, sex or clinical risk group for BNT162b2 or ChAdOx1.

### Anti-S effect on the risk of SARS-Cov-2 infection

9244 individuals were included in an analysis examining the effect of anti-S levels and risk of SARS-Cov-2 infection. Between 14th July 2021 and 30th November 2021, 394 individuals had a breakthrough infection, with 68% of individuals (269/394) experiencing symptoms compatible with COVID infection within 14 days of a positive test. Using quartiles of SARS-CoV-2 anti-S levels, we found the risk of breakthrough infections for the lowest quartile (upper limit 413U/ml) began to diverge from the risk for higher quartiles after around 20 days of follow-up (Fig. 3). We undertook an analysis to examine the total effect of anti-S levels and risk of SARS-CoV-2. In this analysis, each one unit increase in log transformed anti-S levels were associated with a reduced Hazard Ratio (HR) of 0.85 (95% CIs: 0.79–0.92). We found no evidence of an interaction between anti-S levels and second dose vaccine type in our model estimating risk of infection.

### Risk of SARS-CoV-2 infection by vaccine type

We identified 1832 breakthrough infection cases occurring between 1st July 2021 and 30th November 2021 in the test negative case–control study. For each case, four matched controls (7,328) with negative-test results dating between 1st July and 30th November, 2021 (See Supplementary Table S1) were identified. We found an increased risk of a breakthrough infection for those who received the ChAdOx1 compared to those who received BNT162b2 (crude OR: 1.50, 95% CIs:

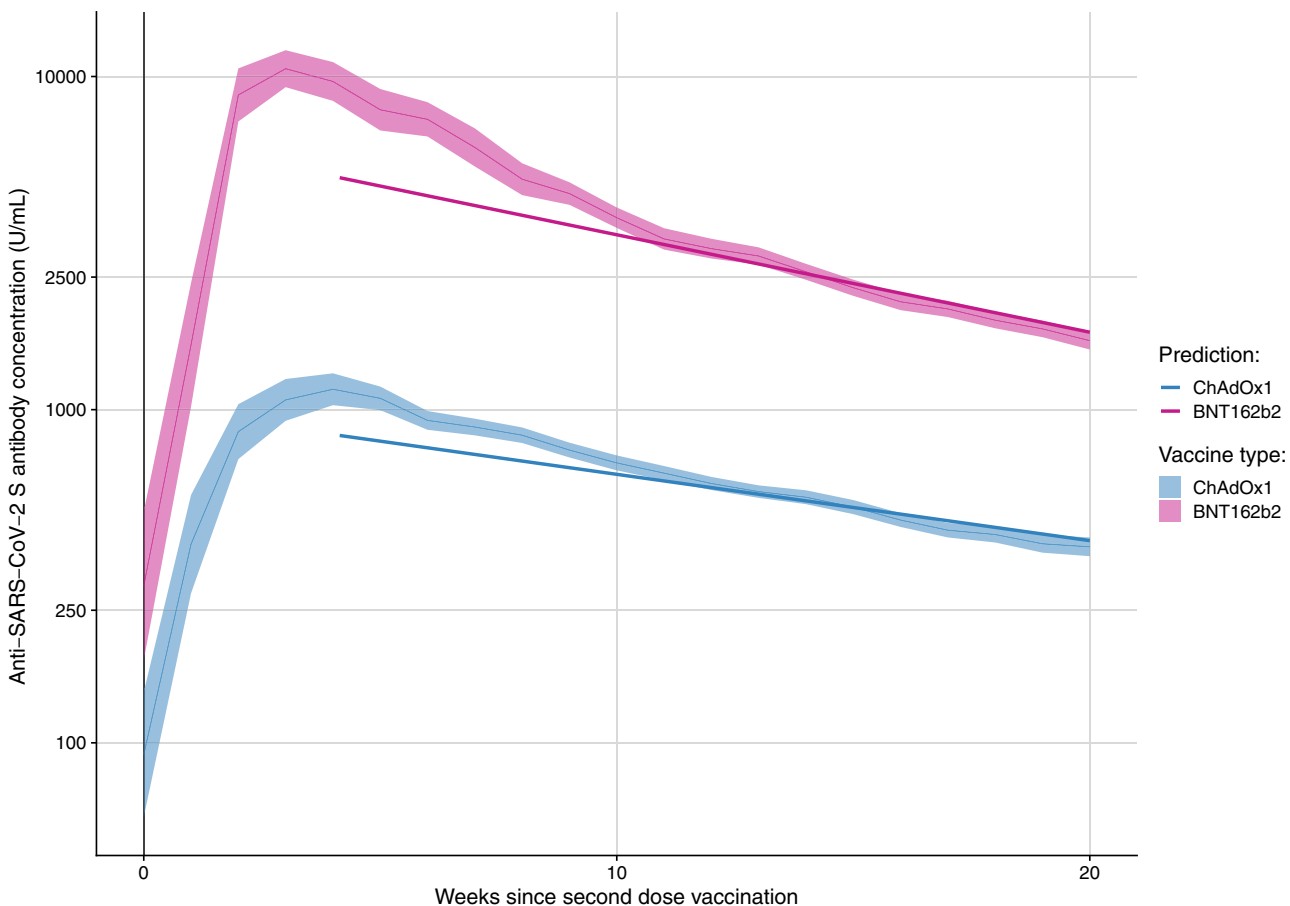

**Fig. 1 | Geometric mean (with 95% confidence intervals) for anti-S samples (U/ml) over time since second dose of vaccination, and levels predicted by a linear mixed effect model amongst N-seronegative individuals by vaccine type.** Wanting anti-S levels followed a mean log-linear decline from 4 weeks after the second dose of vaccination for both vaccine types. The geometric mean of the anti-S samples peaked at 3 weeks after the second dose of vaccine BNT162b2 at 10555 (95% CI: 9291–11992) U/ml and at 4 weeks for ChAdOx1 at 1069 (95% CI: 925–1236) U/ml. At 20 weeks after the second dose of vaccine, the mean anti-S levels were 1611 (95% CI: 1515–1712) U/ml for BNT162b2 and 387 (95% CI: 363–414) U/ml for ChAdOx1.

1.33–1.68, *p* < 0.0001, adjusted OR: 1.64, 95% CIs: 1.45–1.85, *p* < 0.0001).

## Discussion

Our study examined antibody waning in 5834 people that received a second dose of ChAdOx1 and 3363 that received a second dose BNT162b2. Anti-S levels peaked three to four weeks following the second dose of vaccine and the geometric mean of the samples was nine fold higher for BNT162b2 than ChAdOx1. There was substantial waning of anti-S following both vaccines with declines following a log-linear course. We found that higher anti-S levels were associated with a reduced risk of a breakthrough infection, during a period when Delta was the dominant infecting strain in England and Wales. We found no evidence of an interaction between anti-S levels and second dose vaccine type. These data on waning anti-S suggest that those vaccinated with ChAdOx1 are likely at greater risk of breakthrough SARS-CoV-2 infection which we examined further using a test negative case–control analysis from the wider Virus Watch cohort. In this analysis we found that people that had received two doses of ChAdOx1 had 1.64 increased odds of a breakthrough infection compared to those doubly vaccinated with BNT162b2 after we controlled for time since vaccination and a range of demographic and clinical risk factors.

Strengths of our analysis include its community sample design from across England and Wales, with diversity in terms of age, sex and geographical location. We have repeated anti-S levels on a monthly basis from the cohort and levels were measured prior to breakthrough infection with an appropriate time window placed between SARS-Cov-2 infections and anti-S levels to ensure a clear direction of effect with our estimates. The timing of SARS-CoV-2 infections included in the study (14th July 2021 to 30th November 2021) represents a period when Delta was the main circulating strain in England and Wales, which means that our results are relevant to countries that have seen an increase in breakthrough infections due to Delta and where their populations received ChAdOx1 or BNT162b2 vaccines. We have also been able to examine the impact of clinical risk factors on anti-S levels in our analyses and control for these in our comparative evaluation of ChAdOx1 and BNT162b2 vaccines.

In test negative design case–control studies, controls present with symptoms that meet the case definition but test negative for the disease of interest. Compared with traditional case–control studies, this has the advantage that cases and controls have similar participation rates, similar information quality completeness, similar referral areas, and similar evaluation strategies and preferences by doctors[10]. Biases include differences in health-seeking behaviour between cases and controls[11], where the latter can be more sensitive to milder symptoms, potentially leading to relative case underascertainment, and the association of exposure status with being a control[10]. Approximately a quarter of the participants who responded to our occupational survey reported that they were required to test regularly for occupational reasons, which may contribute to the health-seeking behaviour bias. On the other hand, because our study compares the effectiveness of two different vaccines, there is no obvious link between exposure and

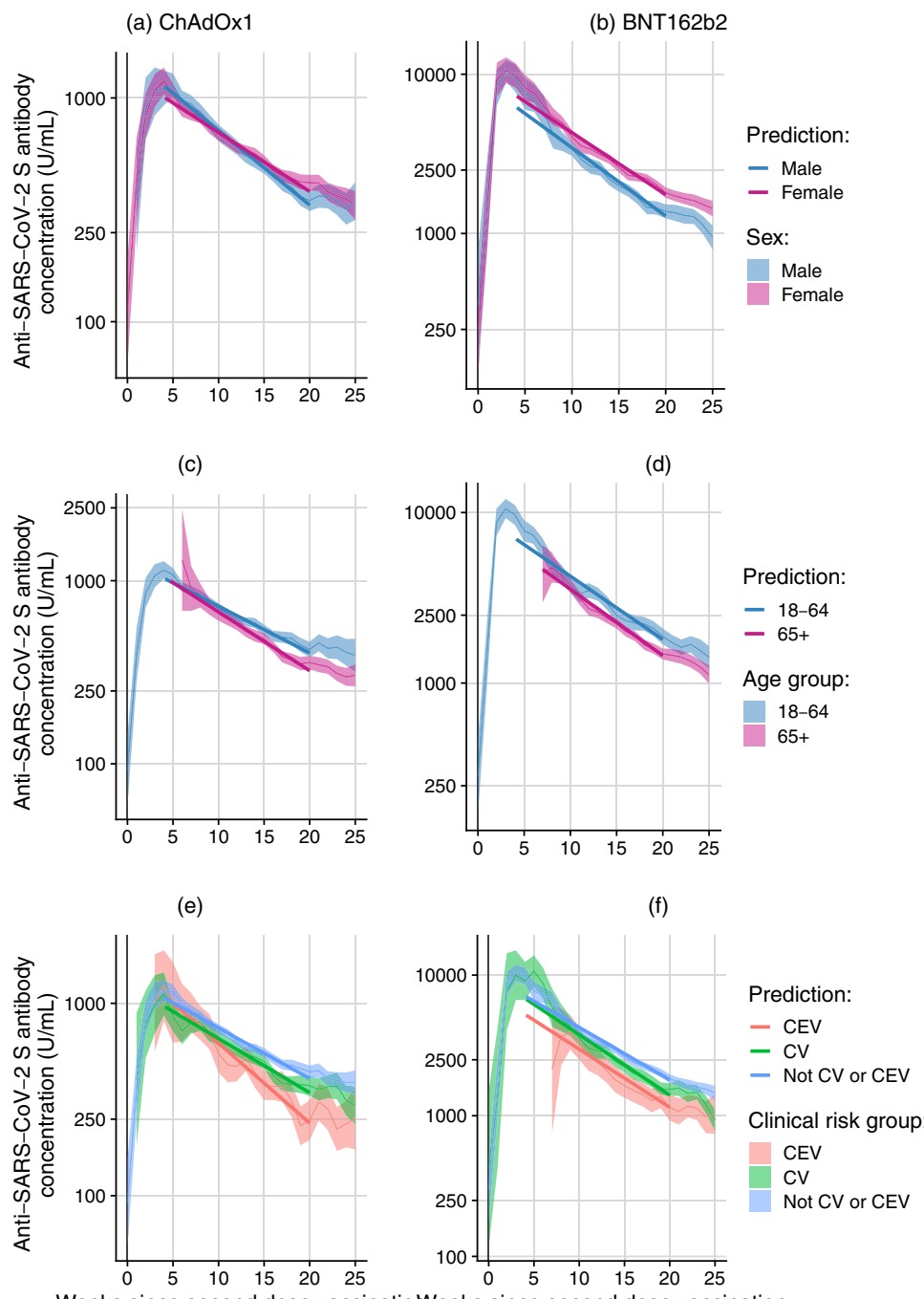

**Fig. 2 | Geometric mean (with 95% confidence intervals) for anti-S samples (U/ml) over time since second dose of vaccination, and levels predicted by a linear mixed effect model amongst N-seronegative individuals by vaccine type and sex, age and clinical risk group.** (Note: Different y-axis scales for BNT162b2 and ChAdOx1). Twenty weeks after the second dose of vaccine, mean BNT162b2 anti-S levels were 1875 (95% CI: 1686–2085) U/ml in 18–64 year olds and 1479 (95% CI: 1373–1593) U/ml in participants over 65 years of age (Fig. 2). Anti-S levels at 20 weeks post second dose of ChAdOx1 were 420 (95% CI: 380–464) U/ml in 18–64 year olds and 356 (95% CI: 328–390) U/ml in participants over 65 years of age. There

was no evidence of a difference in the rates of waning by age, sex or clinical risk group for BNT162b2 or ChAdOx1. ChAdOx1 Oxford, AstraZeneca COVID-19 vaccine. BNT162b2 BioNTech, Pfizer COVID-19 vaccine. Legend description: Panels report on geometric mean anti-S samples (U/ml) over time since second dose of vaccination for Oxford, AstraZeneca by **a** sex, **c** age, **e** clinical risk group and BioNTech, Pfizer by **b** sex, **d** age, **f** clinical risk group. CEV Clinically Extremely Vulnerable, CV Clinically Vulnerable. 18–64 individuals aged 18 to 64. 65+ individuals aged 65 and over. Predictions are the results from linear mixed effect models with a random intercept for age, sex and clinical vulnerability.

control status. As the Virus Watch cohort was not randomly sampled, our measures of absolute risk of infection should be interpreted with caution when generalising to other locations, and at present we do not have data on severe outcomes e.g. hospitalisation or death.

Other longitudinal studies[12] have found that antibody levels 6 months after the second dose of BNT162b2 vaccination decreased on

average to 7% of their peak, and long term follow-up studies of vaccine trial participants have found similar levels of decline and increased risk of breakthrough infection to our findings[13]. Our data are consistent with these findings and provide additional information on peak anti-S levels and trajectories of waning after the second vaccination with both BNT162b2 and ChAdOx1 vaccines. A range of studies using

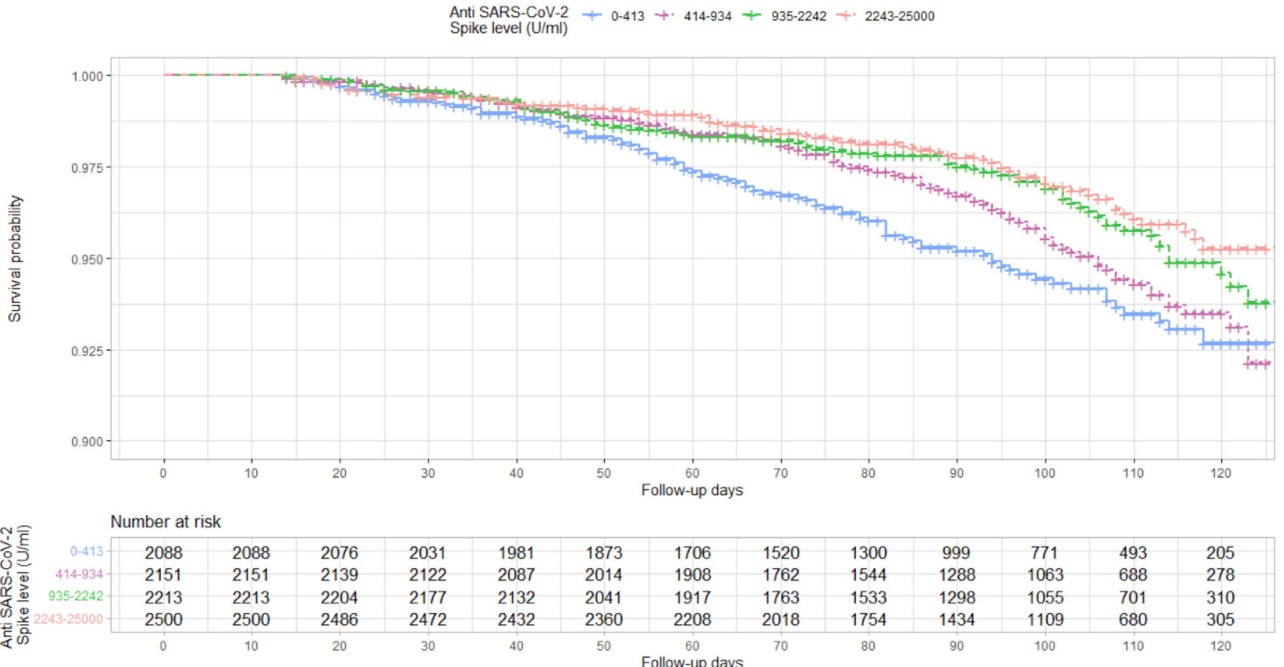

**Fig. 3 | Kaplan–Meier survival curve of risk of breakthrough infection by Spike antibody level in quartiles.** The first anti-S level for participants after their second dose of vaccine and 1st July 2021 was used. The risk of breakthrough infections for the lowest quartile (upper limit 413 U/ml) began to diverge from the risk for higher quartiles after around 20 days of follow-up. The risk of breakthrough infections for the second lowest quartile (414–934 U/ml) began to diverge from the risk for higher quartiles after around 70 days of follow-up.

national surveillance data linked to healthcare records from the UK, Israel and Qatar have been used to monitor BNT162b2 and ChAdOx1 vaccine effectiveness, with a consistent picture that suggests levels of protection to SARS-Cov-2 infection that wanes over time, and is lower for ChAdOx1 than BNT162b2[6,14–17]. A recent study examined the effectiveness of ChAdOx1 versus BNT162b2 COVID-19 vaccines in health and social care workers in England and found no difference in the risk of SARS-CoV-2 infection or COVID-19 disease up to 20 weeks after first dose of vaccination by vaccine type[18]. Our data support this finding up to 20 weeks, but suggest that from this point onwards the risk of breakthrough infection for BNT162b2 is approximately half of ChAdOx1, and our analysis of the total effect of anti-S levels against risk of infection give some indication of the underlying correlates for this increased risk.

Our data provide evidence of the effect of anti-S levels and risk of infection, but other parts of the immune system that are more difficult to measure in large scale population studies are involved in the response to SARS-CoV-2[19]. T-cell-mediated immunity may be particularly important[20] with memory B cells and T cell based immunity providing more stable protection[21,22], providing continued protection against infection and severe disease as antibody responses wane. T-cells have been shown to be important in viral clearance[23] and are correlated with mild disease[24]. Immunological understandings for the less severe disease found in children is still under investigation, but spike-specific T-cell responses were found in seronegative children suggesting a role of cross reactive T-cell responses from seasonal human coronaviruses, and the waning of these cellular responses was found to be slower in children compared to adults[25]. Studies in populations with high exposure to SARS-CoV-2 have demonstrated the presence of virus-specific cellular responses without seroconversion[26,27]. Neutralising responses to SARS-CoV-2 Variants of Concern have been demonstrated to be lowered due to the fact that mutations are often located within the RBD domain of the spike protein which they target[28,29]. However, whilst there remains debate about T-cell immune escape[19], it is unlikely that they will abolish cellular

immune control and vaccines induced T-cell responses have been demonstrated to have broad recognition of SARS-CoV-2 variants[30–33]. Our causal assumptions for this analysis are reported in a directed acyclic graph are reported in Supplementary Fig. S1 and assume that other immune mediators are uncorrelated, which is a limitation of our analysis.

We did not have data on the strain of the infecting virus, but restricted our analyses to time periods when more than 75% of samples tested as part of UK national sequencing data were confirmed to be Delta in all regions in England and Wales. For the majority of our study period (July to mid October 2021) the prevalence of delta was greater than 90% in England. As a result of restricting our analyses to a time period when Delta was the dominant strain, misclassification to other strains is likely to be minimal, but our findings should also be interpreted in light of the generalisability of our results to other strains. Our antibody samples were tested using the Elecsys anti-S and anti-N electro-chemiluminescent immunoassays that targets total antibodies to the S1 subunit of the spike protein and is based upon the ancestral Wuhan Hu-1 strain. The anti-S values presented in this analysis are therefore a proxy for neutralising antibodies and are based upon the immunological relationship between Wuhan Hu-1 and the Delta strain which the majority of participants will have been infected with. The level of protection associated with a particular antibody is likely to vary by the infecting strain, with several analyses suggesting, for example, lower neutralisation efficiency against the beta and omicron variants than Wuhan Hu-1[34].

Overall our data show marked waning of anti-S levels over time since vaccination and provide evidence on the effect of anti-S levels against infection. The fact that anti-S levels start at a much higher level for BNT162b2 than ChAdOx1 means that breakthrough infections are likely to occur significantly earlier in those vaccinated with ChAdOx1 than BNT162b2. The fact that our results showing an increased risk of breakthrough infections for those who were vaccinated with ChAdOx1 compared to BNT162b2 is in line with this hypothesis and is consistent with national and international data showing faster waning of

protection against infection and, to a lesser extent, severe disease, for ChAdOx1 than BNT162b2. Our finding that risk of infection by anti-S level did not interact with vaccine type, but that individuals vaccinated with ChAdOx1 were at higher risk of infection, provides additional support for the use of using anti-S levels for estimating vaccine efficacy.

We anticipate our findings to be useful in the estimation of the protective effect of anti-S levels on risk of infection due to Delta, which occurred prior to immune escape in subsequent variants. The results support the need for booster programmes prioritised to older and clinically vulnerable people who, because of their high risk of severe disease, were prioritised for vaccination in many countries and so have had the longest for antibody levels to decline. In the UK, those aged over 50, those who are clinically vulnerable, and healthcare workers are eligible for boosters from 6 months after their second vaccine. Our results show waning to levels associated with breakthrough infections before this 6 month period for those vaccinated with ChAdOx1 but not for BNT162b2. We also show higher risk of breakthrough infections in those vaccinated with ChAdOx1. Together, these findings suggest that boosting ChAdOx1 earlier than 6 months after the second dose may be advantageous, particularly in those at greatest risk of severe outcomes.

## Methods
The Virus Watch study complies with all relevant ethical regulations and the study protocol has been approved by the Hampstead NHS Health Research Authority Ethics Committee. Ethics approval number −20/HRA/2320. Informed consent was obtained from participants. Virus Watch is a household community cohort of acute respiratory infections in England & Wales that started recruitment in June 2020[35]. To recruit our sample we used a range of methods. We used the Royal Mail Post Office Address File to generate a random list of residential address lists that were sent recruitment postcards, we placed social media adverts on Facebook and Twitter and sent SMS messages and letters to participants from their General Practitioners. Participants were followed-up weekly by email with a link to an illness survey which asked about the presence or absence of symptoms that could indicate COVID-19 disease including respiratory, gastrointestinal and general infection symptoms, in addition to cold-like symptoms such as headache, sore throat and runny nose. The weekly survey was also used to capture SARS-CoV-2 test results received from outside the study (e.g. via the Second Generation Surveillance System system).

### SARS-CoV-2 antibody testing
Nested within the larger Virus Watch study is a sub-cohort of 19,556 adults (aged over 18) participating in antibody testing who completed at-home capillary blood sampling kits sent via post on a monthly basis. We measured antibody titres targeting the spike (S) protein (anti-S) in the context of seronegativity for SARS-CoV-2 anti-Nucleocapsid (anti-N) which is associated with natural infection. Sera were tested using Elecsys anti-S and anti-N electro-chemiluminescent immunoassays (Roche Diagnostics, Basel, Switzerland)[36]. The anti-S assay targets total antibodies to the S1 subunit of the spike protein (range 0.4–25,000 units per mL [U/mL]), whereas the anti-N assay targets total antibodies to the full length nucleocapsid protein, which we took as a proxy for previous SARS-CoV-2 infection (specificity 99.8% [99.3–100])[37]. Individuals were included in this analysis if they underwent antibody testing (anti-N and anti-S) and had a valid result between 1st July 2021 and 30th November 2021. Antibody results were excluded after individuals reported a third booster COVID-19 vaccination.

### SARS-CoV-2 antigen testing
We examined SARS-CoV-2 positive tests confirmed using polymerase chain reaction (PCR) or rapid lateral flow antigen tests (LFD). These positive tests were identified either by participant self-report (1st July

2021 to 30th November 2021) or from linkage of patient demographic characteristics (name, date of birth, address, NHS number) to the national Second Generation Surveillance System for SARS-CoV-2 from 1st July 2021 and 30th November 2021. We did not consider breakthrough infections occurring after third doses.

### Outcomes
We considered two primary outcomes. First, SARS-CoV-2 anti-S titre in the context of seronegativity for SARS-CoV-2 anti-N. Second, SARS-CoV-2 positive tests confirmed using PCR or rapid lateral flow antigen tests. We defined breakthrough SARS-CoV-2 infection as a positive test (PCR or LFD) after being fully vaccinated at least 14 days following the second dose of BNT162b2 or ChAdOx1 and regardless of symptomatology. We only included individuals with anti-S levels measured at least 14 days prior to breakthrough infection to ensure anti-S levels used upon cohort entry were not inflated by early asymptomatic breakthrough infections that develop into symptomatic infections. For this current analysis, we did not examine the presence or absence of symptoms in the context of a positive SARS-CoV-2 test.

### Covariates
We included vaccination status, and vaccine type, collected weekly from 11 January 2021 onwards. Age, sex, ethnicity and geographical region were derived from participants' responses to demographic questions at study baseline. Vaccination status and vaccine type were derived from self-reported data, and data linkage to were identified either by participant self-report or from linkage of patient demographic characteristics (name, date of birth, address, NHS number) to the National Immunisation Management Service (NIMS). We categorised people as clinically vulnerable (CV) or clinically extremely vulnerable (CEV) using our previously reported methods[5]. People were considered clinically extremely vulnerable using criteria set out by Public Health England and the Department of Health and Social Care as part of the guidance for shielding[38], which were adapted in line with clinical variables collected through the Virus Watch baseline survey and a monthly survey. People were categorised as clinically vulnerable (CV) using criteria set out by the Joint Committee on Vaccination and Immunisation on 30 December 2020[39], but excluding those who met the superseding clinically extremely vulnerable criteria (see Supplementary Tables S2, S3). Individuals were classified as not clinically extremely vulnerable, or clinically vulnerable if they did not meet these clinical criteria. We also included individuals with missing data on clinical characteristics as not clinically extremely vulnerable.

### Sample size
The Virus Watch study was initially powered for a testing sub-cohort powered for accurate weekly age-specific disease incidence rates to be measured assuming 20–30% clinical attack rate over 18 weeks. With a clinical attack rate of 30% of whom 20% need hospitalisation and 0.5% die, we expected the following number of outcome events in our testing cohort of 10,000 individuals in study 2: 3000 COVID-19 illnesses, 600 hospitalised cases and 15 deaths.

### Statistical analyses
We undertook three separate analyses. First, to investigate antibody waning we compared anti-S levels for ChAdOx1 and BNT162b2 by time since vaccination, age, sex and clinical vulnerability. For each week since vaccination, we calculated the geometric mean of the anti-S samples and associated 95% confidence intervals. We fit a linear mixed effect model with a random intercept (to account for repeated samples for each participant) to anti-S level data in natural log space 28 days post second vaccination dose (to account for increasing anti-S levels prior to this time point) and predicted the trajectory of waning by vaccine type and ChAdOx1 and BNT162b2. We used the estimated regression coefficient $r$ from the linear model to calculate

the corresponding half-life value associated with anti-S waning, using the formula:

$$t_{1/2} = \frac{ln(0.5)}{r}$$

Equation 1. Formula used for the calculation of half-life.

Second, to investigate the total effect of anti-S level on risk of SARS-CoV-2 infection infection, we undertook a survival analysis using a Kaplan–Meier descriptive analysis in addition to a Cox regression model. For the Kaplan–Meier descriptive analysis we used the first anti-S level after 1st July. Individuals were at risk of infection from 14 days after the first anti-S level and exiting at the first of reported episode of SARS-CoV-2 breakthrough infection or at the end of study follow-up on 30th November 2021 or upon the receipt of the third dose of the vaccine. We created quartiles of anti-S levels based upon samples collected between 1st July 2021–30th July 2021. In our Cox regression model, we included repeated anti-S level measures for individuals between 1st July and 30th November 2021 and individuals exited at the first of reported episode of SARS-CoV-2 infection, upon receiving their third dose of vaccination or at the end of study follow-up on 30th November 2021. In this analysis we included repeated anti-S levels, age, sex, clinical vulnerability and vaccine type. Anti-S level was included as a continuous log-linear variable after checking for evidence of non-linearity ($\chi^2 = 0.00$, $p = 0.84$) by fitting the model with penalised splines for Anti-S level using the pspline function in the survival package in R[40]. We report our causal assumptions for this analysis in a directed acyclic graph (DAG) displayed in Supplementary Fig. S1. Using this DAG we conditioned on clinical vulnerability, age, sex and vaccine type to estimate the total effect of anti-S on risk of SARS-CoV-2 infection.

Third, to investigate how the vaccine type affects the chances of developing a breakthrough infection after the second dose, we undertook a negative-test design case–control study in the full Virus Watch cohort (e.g. not just those participants undertaking monthly antibody testing). Cases were defined as double-vaccinated individuals who had not reported prior infection, reporting a PCR or rapid antigen confirmed infection at least 2 weeks after the second dose of the vaccine and before their third dose, if applicable. Controls were defined as double-vaccinated individuals who had not reported infection throughout the study period, reporting a negative PCR or rapid antigen test result at least 2 weeks after the second dose of the vaccine and prior to their booster dose, if applicable. Four controls were matched to each case based on COVID-19 incidence levels in their lower tier local authority at the time of test. The exposure of interest was the type of vaccine (ChAdOx1 or BNT162b2) and the chosen covariates were age, sex, clinical vulnerability and time since receiving the second vaccine dose. Conditional logistic regression was used to calculate the odds ratio for the exposure of interest in the presence of the covariates.

Data were collected using REDCap 12.4.0 and analyses were conducted in R 4.0.3 and Python 3.6.0. The study is registered with ISRCTN: https://doi.org/10.1186/ISRCTN32077121.

**Reporting summary**
Further information on research design is available in the Nature Research Reporting Summary linked to this article.

## Data availability
The raw data used in this study have been deposited in the ONS Secure Research Service. The data are available under restricted access as they contain sensitive health data. Access can be obtained by ONS Secure Research Service.

## Code availability
Code for these analyses are available from the corresponding author upon request.

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

## Acknowledgements

The research costs for the study have been supported by the MRC Grant Ref: MC_PC 19070 awarded to UCL on 30 March 2020 and MRC Grant Ref: MR/V028375/1 awarded on 17 August 2020. The study also received $15,000 of Facebook advertising credit to support a pilot social media recruitment campaign on 18th August 2020. This study was supported by the Wellcome Trust through a Wellcome Clinical Research Career Development Fellowship to RWA [206602] and a Clinical Ph.D. Fellowship to AA [206441/Z/17/Z] I.B. is supported by an NIHR Academic Clinical Fellowship. S.B. and T.B. are supported by an MRC doctoral studentship (MR/N013867/1). This work was conducted on behalf of the Virus Watch Collaborative. We would like to acknowledge Nick Kennedy at Royal Devon and Exeter NHS Foundation Trust for the idea of plotting geometric mean data as presented in Figs. 1, 2.

## Author contributions

Conceptualisation (R.W.A., A.Y., J.K., V.N., S.B., T.B., W.L.E.F., C.G., P.P., M.S., I.B., A.M.D.N., A.M.J., A.R., A.H.), data curation (R.W.A., E.F., V.N., S.B., T.B., W.L.E.F., C.G., A.M.D.N.), formal analysis (R.W.A., M.E., A.Y.), investigation (All), methodology (All), project administration (J.K.), resources (All), writing—original draft preparation (R.W.A., A.Y., A.H.), writing—review and editing (All).

## Competing interests

ACH serves on the UK New and Emerging Respiratory Virus Threats Advisory Group. A.M.J. is Chair of the Committee for Strategic Coordination for Health of the Public Research. The remaining authors declare no competing interests.
