## [Peer Review File · Nature Communications]

SARS-CoV-2 antibodies and risk of breakthrough infection in
Virus Watch.Editorial Note: This manuscript has been previously reviewed at another journal that is not operating a transparent peer review scheme. This document only contains reviewer comments and rebuttal letters for versions considered at Nature Communications.

Reviewers' Comments:

Reviewer #2:

Remarks to the Author:

The revision was fairly responsive to reviewer comments, including removing the post-selection inference problem and coursening information problem in using a binary marker, adding confounding adjustment, and modifying other statistical analyses to provide improved inferences and interpretable results.

It still remained somewhat vague to me on how to interpret the study endpoint, as a one place the manuscript states infection is measured regardless of symptomology and at another place it is stated that most infections are symptomatic.

Reviewer #3:

Remarks to the Author:

Thanks for the revision. I have a few more comments for consideration.

Summary: provides additional support for the use of using anti-S levels for estimating vaccine efficacy -- suggest changing to "provide additional evidence that anti-S levels may correlate with vaccine efficacy."

Figure 2 bottom row looks funny. You probably need a 3x3 grid here with the right column reserved for legends.

Suggest converting your HRs to log-10 if they are not and describing in terms of 10-fold increases.

Figure 3: suggest adding the time frame for when these antibody measurements are made to the figure title and/or caption. E.g., "post dose 2 antibody levels"

Page 21 of submission packet: there is an error in the parenthetical describing the test of non-linearity -- a hanging, superscript 2.

In the DAG, it's difficult to believe that Anti-S and all "other immune mediators" are uncorrelated. It's a sort of fundamental limitation in the current approaches to COVID-19 correlates analysis that focus exclusively on antibody activity. I don't fault the authors for this; it's a reflection of the current state of the field. Nevertheless, this limitation is probably worth mentioning somewhere.

Reviewer #2:

The revision was fairly responsive to reviewer comments, including removing the post-selection inference problem and coursening information problem in using a binary marker, adding confounding adjustment, and modifying other statistical analyses to provide improved inferences and interpretable results.

It still remained somewhat vague to me on how to interpret the study endpoint, as a one place the manuscript states infection is measured regardless of symptomology and at another place it is stated that most infections are symptomatic.

Our primary outcome was specified a-priori as “We defined breakthrough SARS-CoV-2 infection as a positive test (PCR or LFD) after being fully vaccinated at least 14 days following the second dose of BNT162b2 or ChAdOx1 and regardless of symptomatology.”

In our previous review we were asked by reviewer 2 to “estimate what fraction of infections that are asymptomatic/mild as compared to symptomatic?”. To respond to this comment we added this information in the results section of the updated manuscript which we believe may have now caused this difficulty in interpretation.

We do not feel it is appropriate to make a post-hoc change to our primary outcome, but hopefully this explains why this additional information is now included. We would be happy for the data on symptomatology to be removed from the results section to reduce this confusion if the editors felt it was appropriate to do so.

Reviewer #3:

Thanks for the revision. I have a few more comments for consideration.

Summary: provides additional support for the use of using anti-S levels for estimating vaccine efficacy -- suggest changing to "provide additional evidence that anti-S levels may correlate with vaccine efficacy."

Figure 2 bottom row looks funny. You probably need a 3x3 grid here with the right column reserved for legends.

We agree and have edited Figure 2 and believe we have now resolved this issue.

Suggest converting your HRs to log-10 if they are not and describing in terms of 10-fold increases.

We have followed what we believe to be the typical convention for reporting anti-s levels and use log-e rather than log-10.

Figure 3: suggest adding the time frame for when these antibody measurements are made to the figure title and/or caption. E.g., "post dose 2 antibody levels"

We have edited the figure title to clarify that “The first anti-S level for participants after their second dose of vaccine and 1st July 2021 was used. ”

Page 21 of submission packet: there is an error in the parenthetical describing the test of non-linearity -- a hanging, superscript 2.

We hope that this error is now fixed in the updated pdf submission.

In the DAG, it's difficult to believe that Anti-S and all "other immune mediators" are uncorrelated. It's a sort of fundamental limitation in the current approaches to COVID-19 correlates analysis that focus exclusively on antibody activity. I don't fault the authors for this; it's a reflection of the current state of the field. Nevertheless, this limitation is probably worth mentioning somewhere.

We now note this limitation in the discussion section as suggested by the reviewer.